# CRISPR-Cpf1 mediates efficient homology-directed repair and temperature-controlled genome editing

Miguel A. Moreno-Mateos[1], Juan P. Fernandez[1], Romain Rouet[2,3], Charles E. Vejnar[1], Maura A. Lane[1,4], Emily Mis[1,4], Mustafa K. Khokha[1,4], Jennifer A. Doudna [2,3,5,6,7,8] & Antonio J. Giraldez[1,9,10]

Cpf1 is a novel class of CRISPR-Cas DNA endonucleases, with a wide range of activity across different eukaryotic systems. Yet, the underlying determinants of this variability are poorly understood. Here, we demonstrate that LbCpf1, but not AsCpf1, ribonucleoprotein complexes allow efficient mutagenesis in zebrafish and *Xenopus*. We show that temperature modulates Cpf1 activity by controlling its ability to access genomic DNA. This effect is stronger on AsCpf1, explaining its lower efficiency in ectothermic organisms. We capitalize on this property to show that temporal control of the temperature allows post-translational modulation of Cpf1-mediated genome editing. Finally, we determine that LbCpf1 significantly increases homology-directed repair in zebrafish, improving current approaches for targeted DNA integration in the genome. Together, we provide a molecular understanding of Cpf1 activity in vivo and establish Cpf1 as an efficient and inducible genome engineering tool across ectothermic species.

[1] Department of Genetics, Yale University School of Medicine, New Haven, CT 06510, USA. [2] Department of Molecular and Cell Biology, University of California, Berkeley, CA 94720, USA. [3] California Institute for Quantitative Biosciences, University of California, Berkeley, CA 94720, USA. [4] Department of Pediatrics, Yale University School of Medicine, New Haven, CT 06520, USA. [5] Department of Chemistry, University of California, Berkeley, CA 94720, USA. [6] Innovative Genomics Initiative, University of California, Berkeley, CA 94720, USA. [7] MBIB Division, Lawrence Berkeley National Laboratory, Berkeley, CA 94720, USA. [8] Howard Hughes Medical Institute, University of California, Berkeley, CA 94720, USA. [9] Yale Stem Cell Center, Yale University School of Medicine, New Haven, CT 06510, USA. [10] Yale Cancer Center, Yale University School of Medicine, New Haven, CT 06510, USA. Juan P. Fernandez and Romain Rouet contributed equally to this work. Correspondence and requests for materials should be addressed to M.A.M.-M. (email: moreno.mateos.ma@gmail.com) or to A.J.G. (email: antonio.giraldez@yale.edu)

Cpf1 is a newly discovered class 2/type V CRISPR-Cas DNA endonuclease[1] that displays a range of activity across different systems. Thus far, two Cpf1 proteins have been used for genome editing in mammalian cells, AsCpf1 and LbCpf1, which are derived from *Acidaminococcus sp BV3L6* and *Lachnospiraceae bacterium ND2006*, respectively[1]. CRISPR-Cpf1 presents several advantages for genome engineering, including (i) extended target recognition in T-rich sequences (PAM 5′TTTV)[2], such as non-coding RNAs, 5′ and 3′ UTRs (Supplementary Fig. 1a and Supplementary Data 1), (ii) high specificity in mammalian cells[3, 4], and (iii) shorter crRNA (~ 43 nucleotides (nt)), facilitating in vitro synthesis (Supplementary Figs. 1b–e). CRISPR-Cpf1 has been efficiently used to generate targeted mutations in mice[5–7]; however, AsCpf1 shows lower activity in *Drosophila* and plants, hindering the broad application of Cpf1 across different model systems[8–11]. Here, we investigate the underlying determinants of these differences and whether these insights can be exploited to optimize this method across species and modulate the mutagenic activity of Cpf1 in vivo.

In this study, we characterize and optimize the CRISPR-Cpf1 genome editing system in zebrafish (*Danio rerio*) and *Xenopus tropicalis*. We demonstrate that recombinant Cpf1 protein allows Cpf1-mediated genome editing. In contrast, delayed protein expression from mRNA-encoded Cpf1 results in rapid degradation of unprotected crRNAs in vivo. We show that temperature influences Cpf1 activity in vivo, affecting AsCpf1 more markedly, which explains its lower activity in ectothermic organisms such as zebrafish, *Xenopus*, *Drosophila*[10], and plants[8, 9, 11]. We capitalize on these differences to develop a method that provides temporal control of Cpf1-mediated mutagenesis, resulting in different onset and size of mutant clones. Finally, we show that CRISPR-LbCpf1, together with a single-stranded DNA (ssDNA) donor, significantly increases the efficiency of homology-directed repair (HDR) in zebrafish when compared to CRISPR-Cas9. All together, these results provide a highly efficient and inducible genome engineering system in ectothermic organisms.

## Results

### LbCpf1–crRNA RNP complexes provide robust genome editing.
To implement Cpf1-mediated genome editing, we compared the activity of recombinant proteins and mRNAs encoding codon-optimized AsCpf1 or LbCpf1[1] injected in one-cell-stage zebrafish embryos. We used these with either a pool of three crRNAs targeting *slc45a2* (albino) or *tyr* (tyrosinase), which are genes involved in pigmentation. We observed that ribonucleoprotein (RNP) complexes showed a dramatic increase in activity compared to mRNA delivery of Cpf1 (Fig. 1). While AsCpf1 and LbCpf1 mRNAs were efficiently translated and the proteins were detected by western blot analysis (Supplementary Fig. 2a), different crRNAs were rapidly degraded after mRNA injection (Supplementary Figs. 2b, c). Next, we tested whether structured precursor-crRNAs (pre-crRNAs) result in more stable crRNA, after Cpf1 processing (Supplementary Fig. 2d)[12]. However, pre-crRNAs did not increase mutagenic activity nor the stability of the crRNA (Supplementary Figs. 2e, f). Interestingly, we observed a significant increase in crRNA half-life (Supplementary Figs. 3a, b) when Cpf1–crRNA RNP complexes were pre-assembled before injection, suggesting that Cpf1 protein protects crRNAs from rapid degradation in vivo.

Analysis of AsCpf1 and LbCpf1 RNP complexes revealed that both cleaved DNA in vitro (Supplementary Figs. 3c, e). However, only LbCpf1 induced efficient mutagenesis in zebrafish (Figs. 1d–g, Supplementary Figs. 3d and 4). Individual crRNA-LbCpf1 injections targeting the same locus showed differential activity in vivo (Supplementary Fig. 3f) as previously described in human cells[2]. Notably, co-injection of multiple crRNAs did not

synergistically increase LbCpf1 activity, suggesting that the activity was mostly driven by the most efficient crRNA in the pool (Supplementary Fig. 3f). In contrast to LbCpf1, AsCpf1-showed very low mutagenic activity (Figs. 1e, g). Similar results were obtained when targeting the same loci in *Xenopus tropicalis* using a pool of three crRNAs per locus (Supplementary Figs. 4 and 5). To estimate the rate of germline transmission for each locus, we quantified the number of albino or *tyrosinase* homozygous loss-of-function mutants in the offspring of the zebrafish embryos injected with either AsCpf1–crRNA or LbCpf1–crRNA RNP complexes. We obtained ~88% and ~99% for alb and tyr, respectively, from LbCpf1-injected fish (Supplementary Figs. 3g, h) demonstrating a high level of mutagenesis in the germ cells. In contrast, incrossing AsCpf1-injected fish provided a low rate of homozygous loss-of-function mutants (Supplementary Figs. 3g, h), consistent with the low mutagenic efficiency of AsCpf1. Finally, we tested the off-target activity of two highly efficient crRNAs at their predicted off-targets with ≤5 mismatches (Supplementary Figs. 3f, 6, and 10a, c, Supplementary Data 2 and 3). While we found a very high level of mutagenesis on the target sites, we did not detect any off-target effect (Supplementary Fig. 6). These results are consistent with other studies detecting no off-target cleavage when using RNP[3].

Collectively, these results demonstrate that, in contrast to LbCpf1 mRNA injection, delivery of pre-assembled LbCpf1–crRNA RNP complexes provides a robust and specific genome editing system in zebrafish and *X. tropicalis*.

### Temperature modulates Cpf1 activity in vitro and in vivo.
While AsCpf1 efficiently functions in vitro and in mammalian cells at 37 °C[1, 5–7], its activity is dramatically reduced in zebrafish, *X. tropicalis*, *Drosophila*[10], and plants[8, 9, 11], which develop below 28 °C. Thus, we hypothesized that temperature may impact AsCpf1 activity in vivo. Consistently, we observed that AsCpf1 is less active than LbCpf1 at 25 and 28 °C, although both proteins show comparable cleavage activity at 37 °C in vitro (Supplementary Figs. 5d and 7a).

To determine whether temperature modulates Cpf1 activity in vivo, we compared the mutagenic activity of AsCpf1-injected and LbCpf1-injected embryos raised at different temperatures. We observed that a brief incubation of embryos at 34 °C post-injection significantly increased AsCpf1-mediated gene editing of *tyr* and *slc45a2* when using pre-crRNAs (Figs. 2a–c and Supplementary Fig. 7b), measured by an increase in the number of mosaic mutant embryos and the severity of the phenotype (Figs. 2b, c and Supplementary Fig. 7b). Similarly, LbCpf1 activity increased at higher temperatures, generating ~70% albino-like and ~100% *tyrosinase*-like mutants (Figs. 2d, e and Supplementary Figs. 7b, c). Higher temperatures also increased Cpf1 activity in *X. tropicalis* (Supplementary Fig. 7d), indicating that this effect is observed across different ectothermic organisms. In contrast to Cpf1, higher temperatures did not modulate the activity of SpCas9–sgRNA RNP complexes in vivo or in vitro (Supplementary Fig. 8). All together, these results suggest that temperature specifically influences Cpf1 function in vitro and in vivo. This effect is stronger in AsCpf1, thus explaining the lower efficiency of this protein in ectothermic organisms.

Next, we asked whether temperature control of AsCpf1 activity could be used to modulate genome editing over time. First, we observed that longer incubations at 34 °C increased the number of mutant cells and further improved AsCpf1 efficiency in vivo to a level that was comparable to LbCpf1 at 28 °C (Fig. 2f and Supplementary Fig. 9). This suggests that the crRNA–Cpf1 complex is still active and able to mutagenize the genome later in development, allowing for potential on–off modulation of the

mutagenic activity with temperature over time. To test this possibility, we compared the mutagenic activity of AsCpf1 at 28 °C to embryos incubated at 34 °C at 8–24 h post fertilization (hpf) or 24–48 hpf. In both cases, we observed an increase in the severity of the phenotype and the extent of mutant cells,

suggesting that modulating AsCpf1 activity over time can control the onset of mutagenesis and the number of independent mutant events generated in vivo (Fig. 2g). Together, these results suggest that temperature sensitivity of AsCpf1 activity can be used to modulate genome editing over time.

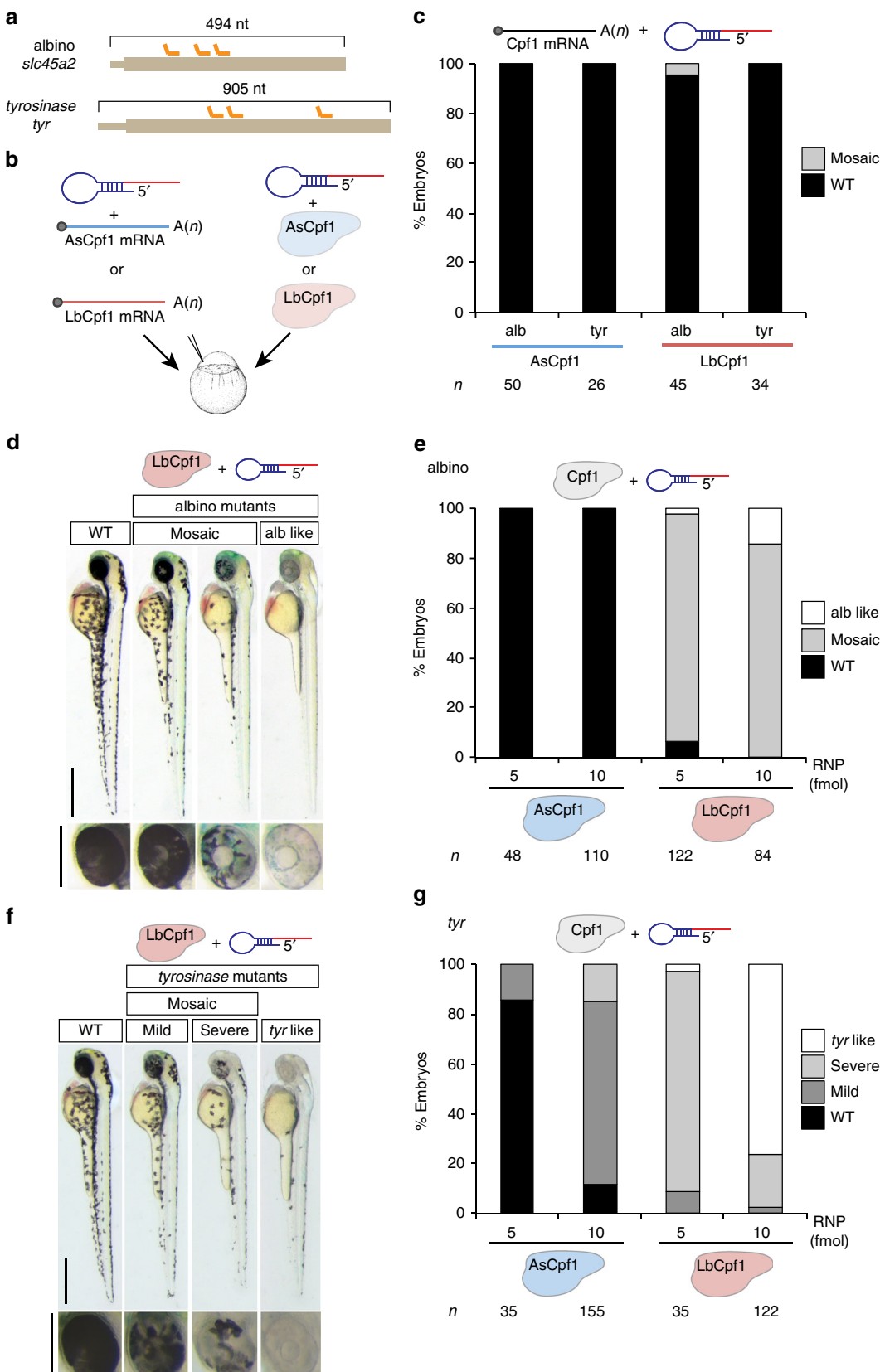

On the basis of the differential activity of Cpf1, we hypothesized that temperature may control Cpf1 endonuclease activity and/or accessibility to the genomic DNA. Previous studies have shown that targeting catalytically dead SpCas9 (SpdCas9) to a genomic locus improves accessibility to the flanking regions of the DNA. This method, called proxy-CRISPR technique, restores the activity of inactive Cpf1 and Cas9 orthologs in mammalian cells[13]. To understand the molecular basis of the temperature modulation of Cpf1 activity, we delivered a proxy-CRISPR adjacent (29–32 nt) to each Cpf1-targeted loci. We observed that proxy-CRISPR enhanced LbCpf1 (Figs. 3a, d, e) and AsCpf1 activity (Figs. 3a–c) at 28 °C, suggesting that temperature likely influences the competence of Cpf1 to access or unwind genomic DNA.

**LbCpf1 enhances homology-directed repair in zebrafish**. Cas9-mediated HDR is low and highly variable in zebrafish[14–16]. One potential drawback of SpCas9 is that SpCas9-induced indels can prevent recurrent cleavage of the target DNA by generating base-pairing mismatches between the "seed" sequence of the target and the sgRNA (Supplementary Fig. 1e)[17]. In contrast, Cpf1 induces a double-strand break (DSB) ~18 nt away from the PAM sequence (Supplementary Fig. 1c)[1, 18]. Thus, we hypothesized that repeated cleavages without destroying the target site may increase the window of opportunity to repair DSB through HDR. Encouraged by the high activity of LbCpf1 in zebrafish, we tested the capability of this endonuclease to facilitate HDR-mediated DNA integration in zebrafish and compared it to SpCas9 across four different loci (Fig. 4, Supplementary Figs. 10 and 11). To this end, we used 13 different ssDNA donors, with a similar structure to those previously tested to optimize Cas9-mediated HDR in cell culture[19]. The main features tested included single-strand DNA donors that were (i) centered on the 3′-end of the DSB, (ii) complementary to either the target or the non-target strand (which contains the PAM sequence), and iii) with different homology arm lengths (Figs. 4a–c, Supplementary Figs. 10d–f and 11a, d, f, i). We observed that LbCpf1 significantly improves HDR in zebrafish compared to SpCas9 in two out of four tested loci, increasing it up to ~4-fold (as average of percentage of HDR per embryo) when compared to an optimized ssDNA donor for SpCas9 (Fig. 4d, Supplementary Figs. 10g and 11e, j)[19]. LbCpf1 and SpCas9 showed similar efficiency inducing HDR in the other two loci (Supplementary Fig. 11). In contrast to SpCas9, Cpf1 induces the highest HDR rate when the single-stranded donor DNA is complementary to the target strand and presents a longer homology arm proximal to the PAM (Figs. 4c, d, Supplementary Figs. 10f, g and 11d, e). Altogether, these results suggest that LbCpf1 activity in combination with optimized ssDNA donors can be used not only as a complement to SpCas9 but also as an improved alternative for HDR approaches.

## Discussion

In this study, we optimize the CRISPR-Cpf1 system and hence broaden the genome editing toolbox available to efficiently generate mutations in non-mammalian vertebrates. In particular, we demonstrate that (i) LbCpf1 RNP complexes efficiently mutagenize the genomes of zebrafish and *X. tropicalis*; (ii) AsCpf1 activity is lower at 28 °C and regulated by temperature, providing a method to modulate mutagenic activity over time, and (iii) in the absence of Cpf1 protein, crRNAs are unstable and rapidly degraded in vivo, explaining the lack of activity when Cpf1 mRNA is injected in zebrafish[7]. Interestingly, previous studies have shown reduced activity of AsCpf1 in *Drosophila*[10] and also across various plant species[8, 9, 11], whereas LbCpf1 is more efficient. Our findings provide a context to understand how AsCpf1 activity is reduced in ectothermic organisms. We demonstrate that the lower activity of AsCpf1 at 28 °C is recovered when SpdCas9 facilitates the genomic DNA accessibility or when zebrafish embryos are incubated at 34 °C. While the lower ability of Cpf1 to access genomic DNA is likely one of the reasons for its decreased endonuclease activity at lower temperatures, we cannot rule out other effects since its in vitro activity is also hampered at temperatures below 37 °C, especially for AsCpf1. Interestingly, this temperature control can be useful to address later mutant phenotypes for genes that function at different developmental stages. Overall, our results suggest that LbCpf1 is more suitable as a constitutive nuclease in ectothermic animals, while AsCpf1 allows temperature and temporal control of the nuclease activity during development.

Moreover, we demonstrate that LbCpf1 achieves higher HDR rates than SpCas9 in zebrafish, likely by allowing repeated cleavages before indel mutations terminate targeting[1], supported by the longer deletions caused by Cpf1 compared to Cas9[3, 10, 20] (Supplementary Figs. 4 and 6b). Repeated cleavages would indeed increase the window of opportunity to repair DSB through HDR instead of alternative end joining, which is the main repair pathway during zebrafish development[16]. In addition, it is also possible that the different DNA lesions induced by Cpf1 and Cas9 may affect the repair process[21]. Interestingly, our results indicate that ssDNA donors complementary to the target sequence increase LbCpf1-mediated HDR. This is in contrast to SpCas9, where previous studies have shown that the highest HDR efficiency is achieved with asymmetric ssDNA donors complementary to the non-target strand[19]. Furthermore, asymmetric ssDNA donors with longer homology arms spanning the PAM sequence improved HDR when using LbCpf1, which suggests that Cpf1 may first release the PAM-proximal target strand, becoming available for HDR. Overall, we provide insights into the molecular mechanisms used by distinct endonucleases to release one DNA strand and enable subsequent repair in vivo.

Finally, we have developed an online resource tool to predict all potential Cpf1 targets and off-targets across *C. elegans*, sea urchin,

---

**Fig. 1** LbCpf1–crRNA RNP complexes are a robust genome editing system in zebrafish. **a** Diagram illustrating three crRNAs (orange) targeting *slc45a2* and *tyr* exon 1 in zebrafish. **b** Schematic illustrating the experimental set-up to analyze CRISPR-Cpf1-mediated mutations in zebrafish. Three crRNAs (**a**) were either mixed with mRNA coding for AsCpf1 or LbCpf1 or assembled into RNP complexes with their corresponding purified proteins and injected in one-cell-stage embryos. **c** Phenotypic evaluation of crRNAs (30 pg/crRNA) and mRNA (100 pg) injections. Stacked barplots showing the percentage of mosaic (gray) and phenotypically wild-type (WT; black) embryos 48 h post fertilization (hpf) after injection. **d** Phenotypes obtained after the injection of the LbCpf1–crRNA RNP complexes targeting *slc45a2* showing different levels of mosaicism compared to the WT. Lateral views (scale bar, 0.5 mm) and insets of the eyes (scale bar, 0.25 mm) of 48 hpf embryos are shown. **e** Phenotypic evaluation of Cpf1–crRNA RNP complexes injections targeting *slc45a2* (albino). Stacked barplots showing the percentage of alb-like (white), mosaic (gray), and phenotypically WT (black) embryos 48 hpf after injection using different amounts (fmol) of RNP complexes. Number of embryos evaluated (*n*) is shown for each condition. **f** Phenotypes obtained after the injection of the LbCpf1–crRNA RNP complexes targeting *tyr* showing different levels of mosaicism compared to the WT. Lateral views (scale bar, 0.5 mm) and insets of the eyes (scale bar, 0.25 mm) of 48 hpf embryos are shown. **g** Phenotypic evaluation of Cpf1–crRNA RNP complexes targeting *tyrosinase* (*tyr*). Stacked barplots showing the percentage of *tyr*-like (white), severe mutant (light gray), mild mutant (dark gray), and phenotypically WT (black) embryos 48 hpf after injection using different amounts (fmol) of RNP complexes. Number of embryos evaluated (*n*) is shown for each condition

sea anemone, *Drosophila*, zebrafish, medaka, *Xenopus*, chicken, mouse, rat, and human genomes in an updated, publicly available resource CRISPRscan: crisprscan.org. Altogether, this study will guide optimization strategies for the CRISPR-Cpf1 system across ectothermic organisms.

## Methods

**crRNA and sgRNA target site design**. Target sites were designed using an updated version of CRISPRscan (crisprscan.org) tool[20]. sgRNAs (5′GGN18-19GG) and crRNAs (5′TTTVN23) target sites without predicted off-targets were used[2, 22].

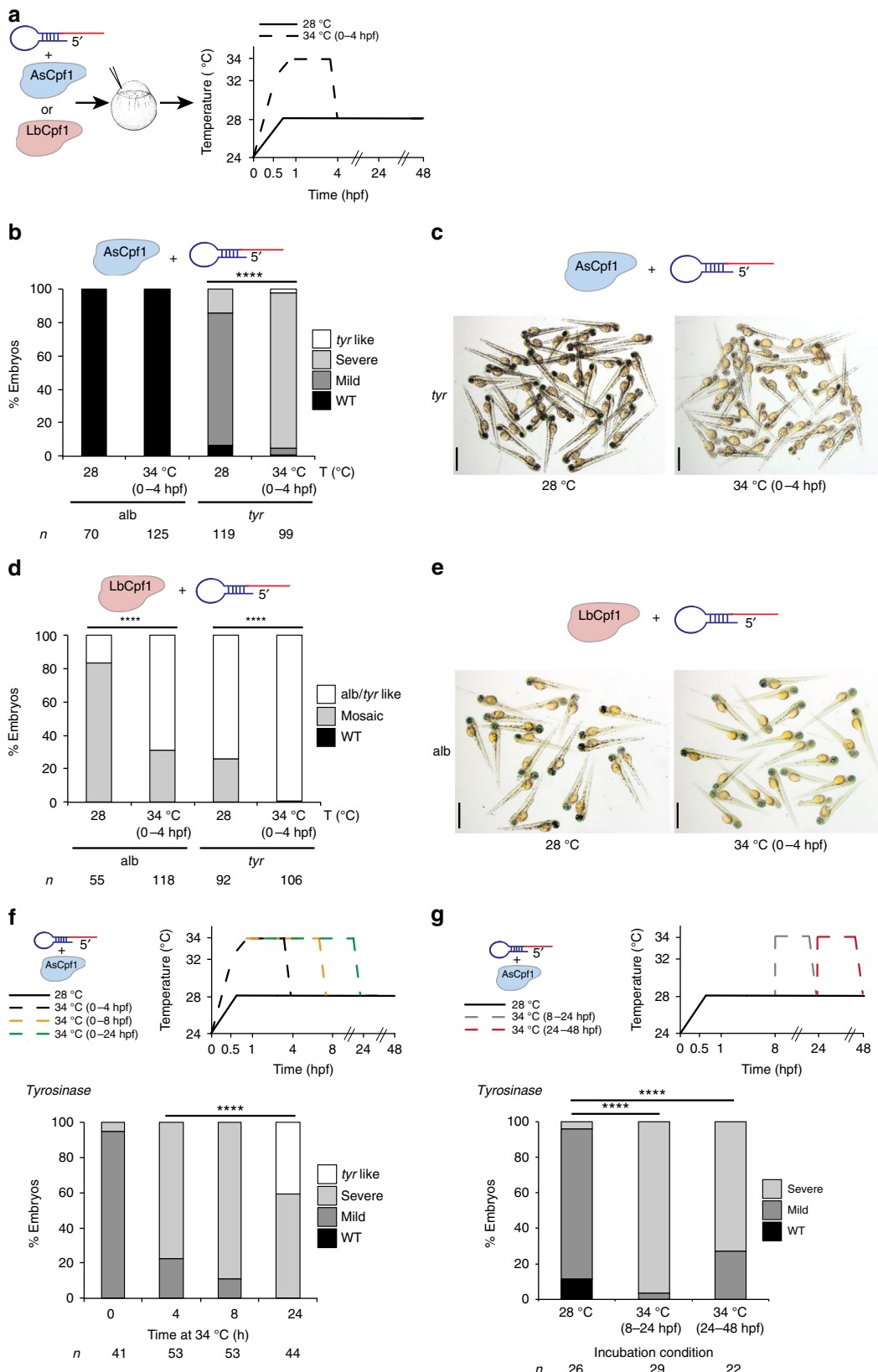

**RNA generation**. crRNA and pre-crRNA DNA template were generated by fill-in PCR (Supplementary Figs. 1d and 2d). A crRNA or pre-crRNA (As/Lb) universal primer (Supplementary Data 1) containing the T7 promoter (5′-TAA-TACGACTCACTATA-3), and the mature crRNA repeat or the complete pre-crRNA repeat for AsCpf1 or LbCpf1 preceded by 5′GG were used in combination with a specific oligo of 43 nt adding the spacer (target-binding sequence) and the repeat sequence (reverse complement orientation). A 65/66 (crRNA) or 80/81 (pre-crRNA) bp PCR product was generated following these conditions: 3 min at 95 °C, 30 cycles of 30 s at 95 °C, 30 s at 52 °C, and 20 s at 72 °C, and a final step at 72 °C for 7 min. PCR products were purified using Qiaquick PCR purification kit (Qiagen) columns and used as template (200–250 ng) for a T7 in vitro transcription reaction (AmpliScribe-T7-Flash transcription kit from Epicenter; 6–7 h of reaction). In vitro transcribed crRNAs were DNAse-treated and precipitated with Sodium Acetate/Ethanol. crRNAs were visualized in a 2% agarose stained by ethidium bromide to check for RNA integrity. sgRNAs were generated by fill-in PCR as previously described[20, 23]. Briefly, a 52-nt oligo containing the T7 promoter, the 20 nt of the specific sgRNA DNA-binding sequence and a constant 15-nt tail for annealing was used in combination with an 80-nt reverse universal oligo to add the sgRNA invariable 3′ end. A 117-bp PCR product was generated, purified and used for a T7 in vitro transcription reaction as described above. sgRNAs and crRNAs targeting *slc45a2* in zebrafish were individually in vitro transcribed. Pre-crRNAs targeting *slc45a2* in zebrafish and crRNAs targeting *slc45a2* and *tyr* in *X. tropicalis* were in vitro transcribed from a pool of PCR templates for each crRNA per gene. Solid-phase extraction-purified crRNAs targeting *tyr* in zebrafish were purchased from Synthego.

A zebrafish codon-optimized AsCpf1 and a human codon-optimized LbCpf1[1] (Addgene; 69988) were PCR-amplified using the following primers: 5′-TTTTccATGGGCACCCAGTTCGAGGGA-3′ and 5′-TTTTCCGCGgtTTATCCGGCGTAATCGGGCACGTC-3′ for Ascpf1 and 5′-ttttGCGGCCGCCACCATGAGCAAGCTGGAGAAGTT-3′ and 5′-ttttgaattcTTAGGCATAGTCGGGGACAT-3′ and 5′-ttttgaattcTTAGGCATAGTCGGGGACAT-3′ for LbCpf1. The following PCR products were then digested with *NcoI* and *SacII* (AsCpf1) or *NotI* and *EcoRI* (LbCpf1), and ligated into the pT3TS-nCas9n[24] and pSP64T plasmids previously digested with these enzymes, respectively. Final constructs were confirmed by sequencing. For making AsCpf1 or LbCpf1 mRNA, the template DNA was linearized using *XbaI* and mRNA was synthesized using the mMachine T3 or SP6 kit (Ambion), respectively. In vitro transcribed mRNAs were DNAse-treated and purified using the RNeasy Mini Kit (Qiagen).

**Protein expression and purification**. *E. coli* codon-optimized SpCas9, SpdCas9 AsCpf1, and LbCpf1 were cloned into pET-based bacterial expression plasmid. AsCpf1 and LbCpf1 expression vectors were deposited to Addgene (#102565 & #102566). Proteins (with N-terminal 6xHis and MBP tags and C-terminal 2xSV40 tags) were expressed in *E. coli* Rosetta 2 in TB media at 16 °C for 18 h following induction with 0.4 mM IPTG. Cells were lysed in 20 mM HEPES pH 7.5, 500 mM KCl, 20 mM imidazole, 5 mM TCEP, 10% glycerol (supplemented with protease inhibitors) by sonication. Proteins in the lysate were first captured onto Ni-NTA resin (Qiagen), washed with lysis buffer, and eluted with 20 mM HEPES pH 7.5, 100 mM KCl, 300 mM imidazole, 5 mM TCEP, 10% glycerol. 6xHis-MBP tag was removed by TEV protease cleavage. Proteins were next dialyzed to 20 mM HEPES pH 7.5, 100 mM KCl, 5 mM TCEP, 10% glycerol and captured onto an ion-exchange column (HiTrap Heparin, GE Healthcare). Proteins were eluted with a linear gradient of 100 mM to 1 M KCl. Finally, size exclusion chromatography (Superdex 200, GE Healthcare) was performed in 20 mM HEPES pH 7.5, 300 mM KCl for Cpf1, or 150 mM KCl for Cas9, 1 mM TCEP, and 10% glycerol. Protein was concentrated and filtered, concentration measured by Abs$_{280\ nm}$ (Nanodrop, Thermo Fisher), and stored at −80 °C.

**RNA and RNP injections**. An aliquot of 100 pg of *cpf1* mRNA and 30 pg of each crRNA or pre-crRNA were injected at the one-cell stage, respectively.

crRNAs or sgRNAs were resuspended in 20 mM HEPES pH 7.5, 1 mM TCEP, 10% glycerol, and 300 or 150 mM KCl, respectively, at 24 μM incubated at 70 °C for 5 min and cooled down to room temperature. Then MgCl$_2$ was added to a final concentration of 1 mM, crRNAs or sgRNAs were incubated at 50 °C for 5 min, cooled down to room temperature and stored at −80 °C. Cpf1–crRNA or Cas9-sgRNA RNP were prepared as follows: Cpf1 or Cas9 were diluted to 20 μM in 20 mM HEPES pH 7.5, 1 mM TCEP, 1 mM MgCl$_2$, 10% glycerol 300 mM or 150 mM KCl, respectively, and 10 μl were added to 10 μl of crRNA/pre-crRNA or sgRNA at 24 μM (Protein-RNA ratio 1:1.2). RNPs (10 μM) were incubated at 37 °C for 10 min and then kept at room temperature before use. RNPs were stored at −80 °C and up to three cycles of freeze–thaw cycles maintained similar efficiency. Two nl (20 fmol), one nl (10 fmol), or 0.5 nl (5 fmol) from 10 μM solution were injected at the one-cell-stage zebrafish embryos. For HDR experiments, 20 (golden) or 10 fmol of RNP complexes and 40 pg of ssDNA donor were injected in one-cell-stage zebrafish embryos.

**LbCpf1 off-target analysis**. Ten injected (LbCpf1–crRNA alb 2 or LbCpf1–crRNA tyr 1, 10 fmol each) or non-injected (as a control) embryos were collected at 28 hpf and DNA was extracted[25] (see below). PCR products of ~100–120 bp were obtained for each of the 15 loci (Supplementary Data 2) using these parameters: 3 min at 95 °C, 35 cycles of 30 s at 95 °C, 30 s at 55 °C, and 30 s at 72 °C, and a final step at 72 °C for 7 min. PCR products were visualized and quantified on agarose gel (Adobe Photoshop). Next, similar amounts of PCR products per gene were pooled and purified using DNA Clean and Concentrator-5 kit with Zymo-Spin IC columns (Zymo Research). Purified amplicons were used to generate DNA libraries and were subjected to deep sequencing (Illumina Hi-Seq sequencer, paired end, with 75-nucleotide reads).

Potential LbCpf1 off-targets with three or more mismatches were detected using CCTop[26]. Detecting potential off-targets requires a comprehensive search of mutations. We optimized the MutSeq pipeline[23] to increase its sensitivity to detect mutations. The heuristics used by short read mappers such as GMAP[27], used in MutSeq version 1, and STAR[28] favor fast mapping of perfect-matching reads and do not guarantee to return alignments for all reads containing complex mutations (i.e., with mixture of mutations, insertions, and deletions). To account for this, we introduced a second pass in the pipeline to produce an alignment with the Smith–Waterman algorithm after the first mapping pass with GMAP or STAR. First, all reads were mapped with STAR allowing soft-clipping and using an index created with on-target and potential off-target sequences, including PCR oligo sequences. Since we guarantee a high-quality alignment with the second pass, we speed up the first pass by replacing GMAP by STAR. Second, perfectly mapped reads were counted as "WT reads" and remaining reads were split into (i) fully aligned reads (no soft-clipping; see options --trim_max and --min_padding) and searched for indels or (ii) re-aligned using the SSW library[29] that implements a vectorized version of the Smith–Waterman algorithm. Lastly, these re-aligned reads were searched for indels in the Smith–Waterman alignment. We restricted the allele search to 20 nt upstream and downstream of LbCpf1 cleavage site (after position 18–23 in target site without including the PAM; see --target option). MutSeq pipeline was started with the following command line options (input folder data contain the reads): mutseq_run.py --path_loci loci.txt --path_data./data --report report.txt --processor 8 --min_allele_reads 10 --cat_cmd zcat --map_soft star --two_pass --target.

File loci.txt (CSV formatted file) is available as Supplementary Data 3. All changes are publicly available in MutSeq pipeline version 1.1 (protocol.crisprscan.org).

**Quantitative RT-PCR**. For crRNA experiment, 0.5 nl of Cpf1 RNP complexes were injected at the one-cell stage and a pool of 10 embryos were collected at 0, 2, and 5 hpf. Total RNA was isolated from embryos injected using TRIzol reagent (Life Technologies). An amount of 1 μg of purified total RNA was then subjected to reverse

**Fig. 2** Temperature is a key factor modulating Cpf1 activity in vitro and in vivo. **a** Schema illustrating different temperature incubations after Cpf1–crRNA RNP complex injections targeting *slc45a2* (alb) and *tyr*. **b** Phenotypic evaluation of AsCpf1–crRNA RNP complex (10 fmol) injections at different temperature incubations (T). Stacked barplots showing the percentage of *tyr*-like (white), severe mutant (light gray), mild mutant (dark gray), and phenotypically WT (black) embryos 48 hpf after injection. Number of embryos evaluated (*n*) is shown for each condition. $\chi^2$-test (****$p < 0.0001$). **c** A representative picture showing 48-hpf-old embryos obtained after AsCpf1–crRNA RNP complex injections targeting *tyr* at different temperature incubations. Scale bar, 1.25 mm. **d** Phenotypic evaluation of LbCpf1–crRNA RNP complex (10 fmol) injections at different temperature incubations (T). Stacked barplots showing the percentage of alb/*tyr*-like (white), mosaic mutants (gray), and phenotypically WT (black) embryos 48 hpf after injection. Number of embryos evaluated (*n*) is shown for each condition. $\chi^2$-test (****$p < 0.0001$). **e** A representative picture showing 48-hpf-old embryos obtained after LbCpf1–crRNA RNP complex injections targeting *slc45a2* at different temperature incubations. Scale bar, 1.25 mm. **f** Schematic illustrating different incubation conditions (0, 4, 8, or 24 h at 34 °C, then at 28 °C) after AsCpf1–crRNA RNP complex (10 fmol) injections targeting *tyr* in zebrafish (top). Phenotypic evaluation of AsCpf1–crRNA RNP complex injections targeting *tyr* in the conditions described above (bottom). Stacked barplots showing the percentage of *tyr*-like (white), severe mutant (light gray), mild mutant (dark gray), and phenotypically WT (black) embryos 48 hpf after injection. Number of embryos evaluated (*n*) is shown for each condition. $\chi^2$-test (****$p < 0.0001$). **g** Schematic illustrating different incubation conditions: 8 h at 28 °C, 16 h at 34 °C, and then 24 h at 28 °C (34 °C 8–24 hpf) or 24 h at 28 °C, then 24 h at 34 °C (34 °C 24–48 hpf) after AsCpf1–crRNA RNP complex (10 fmol) injections targeting *tyr* in zebrafish (top). Phenotypic evaluation of AsCpf1–crRNA RNP complex injections targeting *tyr* in the conditions described above (bottom). Stacked barplots showing the percentage of severe mutant (light gray), mild mutant (dark gray), and phenotypically WT (black) embryos 48 hpf after injection. Number of embryos evaluated (*n*) is shown for each condition. $\chi^2$-test (****$p < 0.0001$)

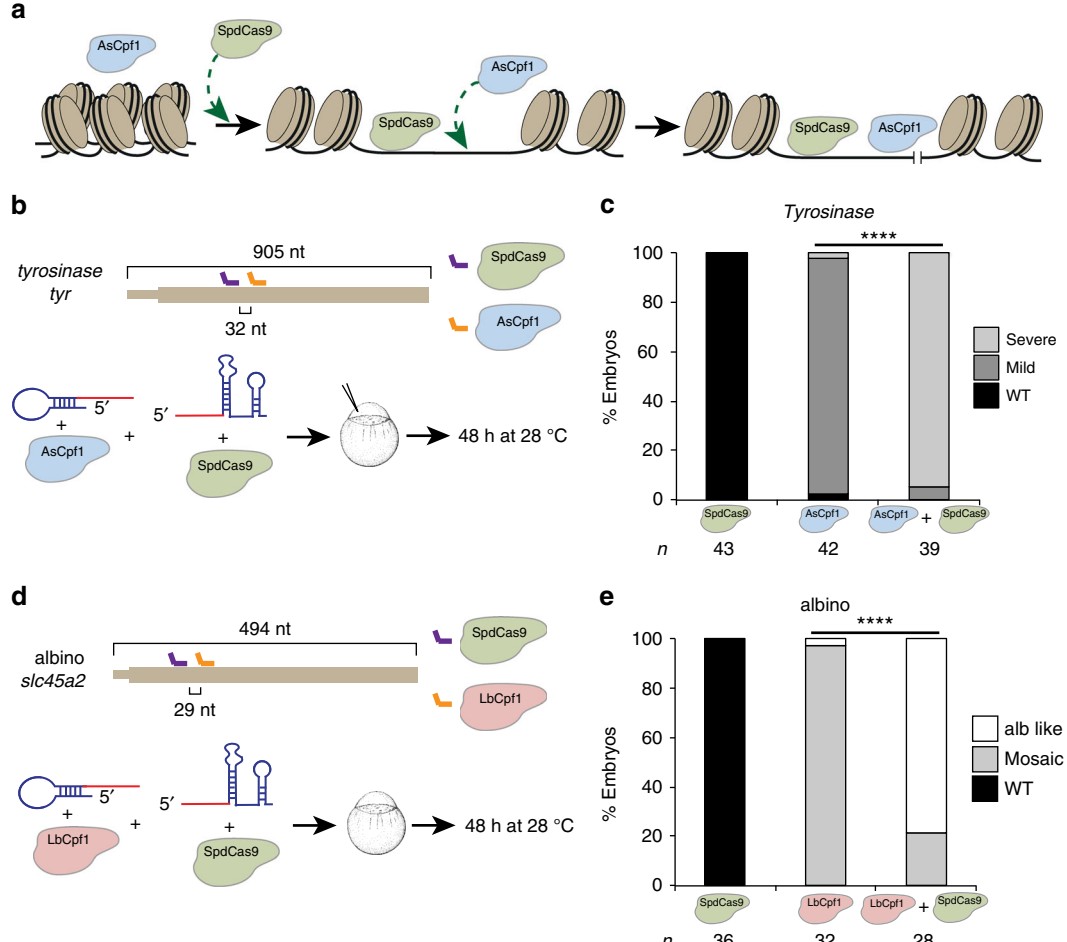

**Fig. 3** Catalytic dead SpCas9 (SpdCas9) proximal targeting increases Cpf1 activity. **a** Schematic diagram of proxy-CRISPR approach. Temperature may control Cpf1 activity to access and/or unwind genomic DNA, impeding AsCpf1 to access the genomic target in vivo at 28 °C. When SpdCas9 binds in the proximity of the AsCpf1 target, it facilitates the availability of AsCpf1 to cleave the inaccessible target. **b** Diagram illustrating crRNA 1 (orange) and a proximal sgRNA (purple) targeting *tyr* exon 1 (Supplementary Fig. 10c and Supplementary Data 1) in zebrafish (top). AsCpf1–crRNA and/or SpdCas9-sgRNA RNP complexes were injected into one-cell-stage embryos and then incubated at 28 °C for 48 h (bottom). **c** Phenotypic evaluation of the experiment described in (**b**). Stacked barplots showing the percentage of severe mutant (light gray), mild mutant (dark gray), and phenotypically WT (black) embryos 48 hpf after injection. Number of embryos evaluated (*n*) is shown for each condition. $\chi^2$-test (****$p < 0.0001$). **d** Diagram illustrating crRNA 2 (orange) and sgRNA 2 (purple) targeting alb exon 1 (Supplementary Data 1, Supplementary Figs. 3f and 8e) in zebrafish (top). LbCpf1–crRNA and/ or SpdCas9-sgRNA RNP complexes were injected into one-cell-stage embryos and then incubated at 28 °C for 48 h (bottom). **e** Phenotypic evaluation of the experiment described in (**d**). Stacked barplots showing the percentage of alb-like (white), mosaic mutants (gray) and phenotypically WT (black) embryos 48 hpf after injection. Number of embryos evaluated (*n*) is shown for each condition. $\chi^2$-test (****$p < 0.0001$)

transcription using the SuperScript® III First Strand Synthesis System (Thermo Fisher Scientific), using random hexamers and a specific primer for each crRNA (Supplementary Fig. 2b and Supplementary Data 1) following the manufacturer's protocol. Five microliters from a 1/50 dilution of the cDNA reaction was used to determine the levels of different crRNAs in a 20 µl reaction containing 1 µl of each oligo crRNA-specific and universal primers or forward and reverse (10 µM; Supplementary Data 1), using Power SYBR Green PCR Master Mix Kit (Applied Biosystems) and a ViiA 7 instrument (Applied Biosystems). PCR cycling profile consisted of incubation at 50 °C for 2 min, followed by a denaturing step at 95 °C for 10 min and 40 cycles at 95 °C for 15 s and 60 °C for 1 min. *taf15* and *cdk2ap2* genes expressed at similar levels during the first 5 hpf were used as normalization controls[30].

For HDR experiments, 28 hpf injected zebrafish embryos were collected for DNA extraction[25]. Briefly, embryos were incubated in 80 µl of 100 mM NaOH at 95 °C for 15 min. Next, 40 µl of Tris-HCl 1 M pH 7.5 was added. Crude DNA extracts were 1/5 diluted (bi-distilled water) and 5 µl were used for qPCR with the corresponding forward and reverse primers (10 µM; Supplementary Data 1 and Supplementary Figs. 10b, h), and using the same conditions described above. % HDR was measured using integration primers and genomic DNA primers (Supplementary Fig. 10; Supplementary Data 1, Fig. 4, and Supplementary Fig. 11). For golden and ntla, specific integration primers and albino genomic DNA primers were used to estimate the amount of genomic DNA (Supplementary Fig. 11 and Supplementary Data 1). To calculate percentage of HDR, standard curves of different genomic qPCR products were performed using known amounts of DNA gBlocks® gene fragments (IDT;

Supplementary Data 1). Then, percentage of HDR per embryo was calculated dividing the amount of integrated DNA by total amount of genomic DNA.

**Western blot**. Ten embryos were collected at 6 hpf and transferred to 200 µl of deyolking buffer for washing (55 mM NaCl, 1.8 mM KCl, and 1.25 mM NaHCO₃). Deyolking buffer was discarded and 200 µl of the same buffer were added to resuspend the embryos by pipetting. The resuspended embryos were incubated at room temperature for 5 min with orbital shaking, and then centrifuged at 300 x*g* for 30 s and washed with 110 mM NaCl, 3.5 mM KCl, 10 mM Tris-HCl pH 7.4, and 2.7 mM CaCl₂. The pellet was resuspended in SDS sample buffer before separation by SDS-PAGE and transferred to PVDF membrane. Anti-HA antibody (Sigma-Aldrich, 11867423001; 1:1000) and rabbit polyclonal Anti-actin antibody (Sigma-Aldrich, A2668; 1:5000) were used according to the manufacturer's instructions. Secondary antibodies were fluorescence-labeled antibodies (alexa fluor 680) from Thermo Fisher (A21057, A10043) and used according to the manufacturer's instructions. Protein bands were visualized using the Odyssey Infrared Imaging System (LI-COR Biosciences, Lincoln, NE, USA).

**In vitro cleavage assays**. The assays were carried out as described in Jinek et al.[31] with minor modifications. Briefly, *slc45a2* and *tyr* targeted region (exon 1) from zebrafish or *X. tropicalis* were amplified by PCR (Supplementary Data 1) and PCR products purified using QIAquick PCR purification kit (Qiagen). An amount 100

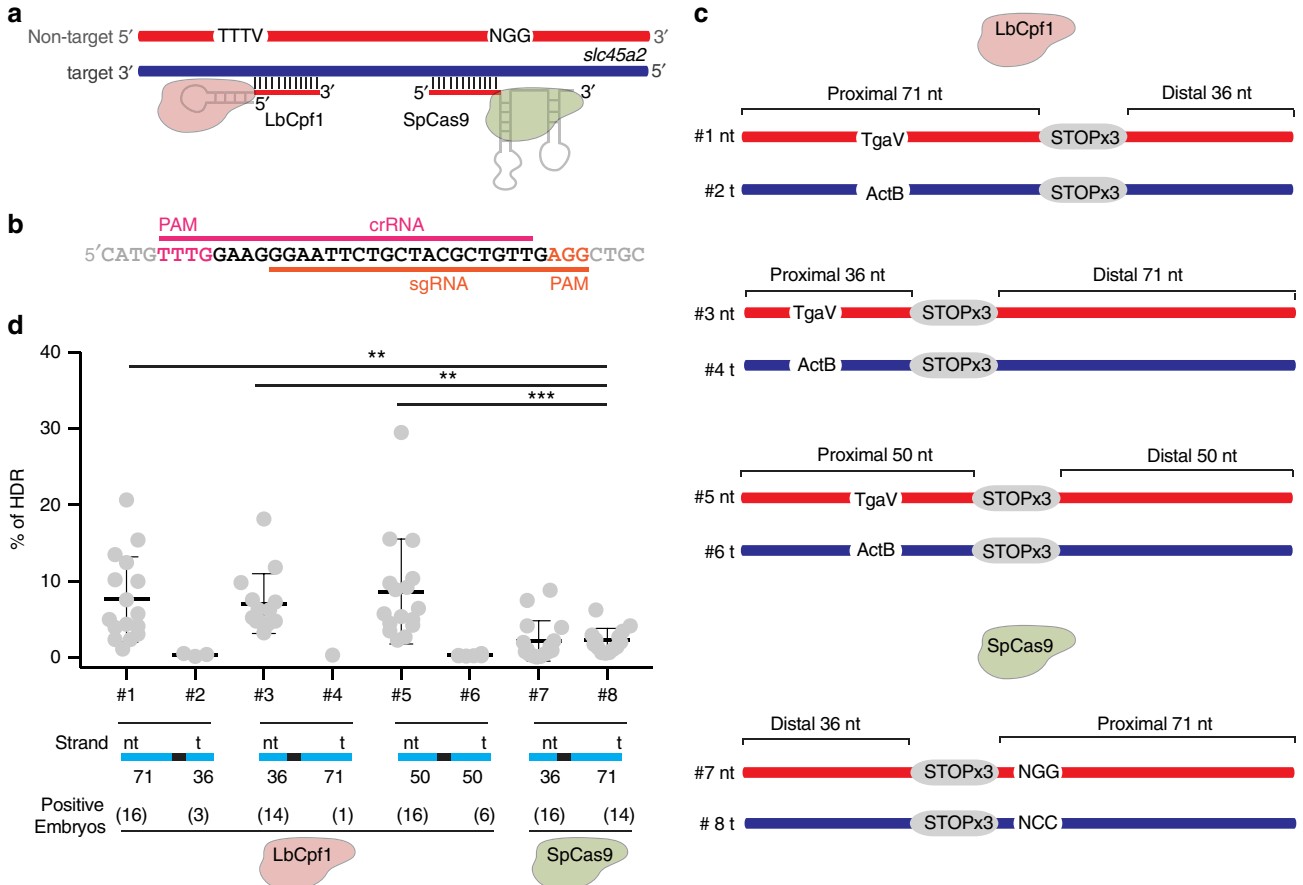

**Fig. 4** LbCpf1-mediated homology-directed repair. **a** Schematic illustrating LbCpf1–crRNA RNP complex and SpCas9–sgRNA RNP complex interaction with their respective DNA targets. crRNA and sgRNA-DNA annealing occurs on the target strand (blue), and PAM sequences are in the non-target strand (red). **b** crRNA (pink line) and sgRNA (orange line) overlapping target sequences in the albino locus used for this analysis. **c** Schema illustrating different donor ssDNA (#1–6) complementary to either the target strand (t) or non-target strand (nt) and with symmetric or asymmetric homology arms used in combination with LbCpf1–crRNA. PAM sequence was modified (TgaV/ActB) to prevent new editing post-HDR. An optimized ssDNA donor (#8) described for SpCas9-induced HDR[19] and its complementary version (#7) were used in combination with SpCas9–sgRNA RNP complexes as references for comparison (bottom). **d** qPCR quantification showing % of HDR from individual embryos when using LbCpf1 and different ssDNA donors in comparison to SpCas9. % of HDR: amount of integrated DNA per total amount of genomic DNA per embryo (see Methods for details). Results are shown as the averages ± S.D. of the means from 16 embryos in two independent experiments (n = 8 embryos per experiment). Positive embryos: number of embryos per condition showing a detectable qPCR amplification signal. The data were analyzed by Kruskal–Wallis test followed by Dunn's post test for significance vs. control condition (#8), **p < 0.01, ***P < 0.001

ng of PCR-purified product of *slc45a2* zebrafish (~0.11 pmol), *tyr* zebrafish (~0.23 pmol), *slc45a2* X. tropicalis (~0.45 pmol), and *tyr* X. tropicalis (~0.38 pmol) were subjected to in vitro cleavage with different concentrations of Cpf1 or SpCas9 RNP complexes in cleavage buffer (20 mM HEPES pH 7.5, 150 mM KCl, 0.5 mM DTT, 0.1 mM EDTA, and 10 mM MgCl$_2$) at different temperatures for 90 min followed by 37 °C for 5 min incubations with proteinase K (20 µg). The reactions were stopped with SDS loading buffer (30% glycerol, 0.6% SDS, and 250 mM EDTA) and loading 1.5% agarose gel stained by ethidium bromide.

**Zebrafish maintenance and image acquisition**. Zebrafish wild-type embryos were obtained from natural mating of TU-AB and TLF strains of mixed ages (5–17 months). Selection of WT or LbCpf1–crRNA RNP-injected mosaic F0 mutant mating pairs was random from a pool of 48 males and 48 females allocated for a given day of the month or random from the F0 mosaic mutant adult fish obtained ~ 3 months after injection, respectively. Fish lines were maintained in accordance with AAALAC research guidelines, under a protocol approved by Yale University Institutional Animal Care and Use Committee.

All experiments were carried out at 28 or 34 °C, temperatures allowing normal development[32].

Embryos were analyzed using a Zeiss Axioimager M1 and Discovery microscopes and photographed with a Zeiss Axiocam digital camera. Images were processed with Zeiss AxioVision 3.0.6.

**Frog husbandry and injections**. X. tropicalis were housed and cared for in Khokha Lab aquatics facility according to the established protocols approved by the Yale Institutional Animal Care and Use Committee.

In vitro fertilization and microinjection were carried out at 23.5 °C, as previously described[33, 34]. Two nanoliters of RNP at 10 µM (20 fmol) were injected into one-cell stage embryo. After injection, embryos were left in 3% Ficoll for 30 min, and then transferred to growing medium (1.1 mM Mg$_2$Cl, 2.2 mM Ca$_2$Cl, 2 mM KCl, 11 mM NaCl, and 5.5 mM HEPES pH 7.4) and incubated at different temperatures allowing normal development[34].

**Statistics**. No statistical methods were used to predetermine sample size. The experiments were not randomized and investigators were not blinded to allocation during experiments and outcome assessment. Bar graphs and scatter plots are represented with S.D. error bars. Unpaired two-tailed *t*-test, unpaired two-tailed Mann–Whitney test, or Kruskal–Wallis test followed by Dunn's post hoc test were performed and *p* values were calculated with Prism (GraphPad Software, La Jolla, CA, USA). $\chi^2$-test of contingency with Yates' correction for continuity was used to compare the results from different injections and/or incubation conditions.

**Data availability**. Input raw reads are publicly accessible in the Sequence Read Archive under SRP117270. All other relevant data are available from corresponding authors upon reasonable request.

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

## Acknowledgements

We thank H. Codore and K. Bishop for technical help; D. Burkhardt and L. Miao for helping with primer design and HDR experiments respectively; J.-D. Beaudoin, V. Tornini, and C. Fellmann for discussions and all the members of the Giraldez laboratory for intellectual and technical support; M. Fernandez-Fuertes for helping with western blots and C. Takacs and V. Tornini for manuscript editing. We thank Synthego for providing synthetized crRNAs, and K. Bilguvar and I. Tikhonova for sequencing support; Programa de Movilidad en Áreas de Investigación priorizadas por la Consejería de Igualdad, Salud y Políticas Sociales de la Junta de Andalucía (M.A.M.-M.), NIH grants R21 HD073768, R01 HD074078, GM103789, GM102251, GM101108, and GM081602 (A.J.G.), The Swiss National Science Foundation grant P2GEP3_148600 (C.E.V.) and R01 HD081379, 4R33HL120783 (E.M., M.A.L., and M.K.K.) supported this work. Research reported in this publication was supported by the National Institute Of General Medical Sciences of the National Institutes of Health under Award Number R35GM122580. The content is solely the responsibility of the authors and does not necessarily represent the official views of the National Institutes of Health. M.K.K. is supported by the Edward Mallinckrodt Jr Foundation. A.J.G. is supported by the HHMI Faculty Scholars program, the March of Dimes, the Yale Scholars Program, and the Whitman fellowship funds provided by E.E. Just, L.B. Lemann, E. Evelyn, and M. Spiegel, the H. Keffer Hartline and E.F. MacNichol Jr at the Marine Biological Laboratory in Woods Hole, M.A. M.K.K. is supported by the Edward Mallinckrodt Jr Foundation. R.R. acknowledges support from the Australian National Health and Medical Research Council for his early career postdoctoral fellowship (APP1090875). J.A.D. is an investigator of the Howard Hughes Medical Institute.

## Author contributions

M.A.M.-M. and A.J.G. conceived the project and M.A.M.-M., R.R. and A.J.G. designed the research. M.A.M.-M. performed all zebrafish experiments and in vitro assays, R.R. purified and provided recombinant proteins, J.P.F. designed and performed HDR experiments with M.A.M.-M. C.E.V. carried out off-target analysis, developed crisprscan. org and help with statistical analysis. E.M. and M.A.L. carried out *Xenopus* injections and phenotype analyses. M.A.M.-M., R.R., J.P.F., A.J.G. and J.A.D. performed data analysis and M.A.M.-M., R.R. and A.J.G. wrote the manuscript with input from the other authors. M.K.K. provided reagents and materials. All authors reviewed and approved the manuscript.
