## [Peer Review File · Nature Communications]

Reviewers' comments:

Reviewer #1 (Remarks to the Author):

Moreno-Mateos et al. compare nuclease efficiency with LbCpf1 and AsCpf1 in zebrafish and find that brief pulses of elevated temperature enhance KO efficiency. They also compare Cpf1 with SpCas9 and find increased ssODN-based HR using Cpf1. Overall, I think these optimized parameters for efficient editing in zebrafish will be broadly useful to the community but, to ensure that this is the case, a few issues need to be addressed

Comments:

* Since the authors are advocating use of Cpf1 instead of Cas9 (which works very well in zebrafish), it is necessary to compare the new CRISPRs with Cas9. For Fig. 1 and 2, the authors should provide parallel experiments with Cas9 using the same conditions/temperature changes. Does LbCpf1 outperform Cas9 for routine KO?

* What about off-target effects? Is Cpf1 superior to Cas9 or similar in this regard (i.e. number of off-target modifications)? Even if there is greater on-target modification with Cpf1, perhaps there is greater off-target modification also and this would recommend against Cpf1 (vs. Cas9).

* The PAM for Cpf1 has now been shown to be TTTV not TTTN. Did the authors use TTTT sequences and, if so, can they comment on differences with TTTV PAMs?

* Although Figure 4 looks very promising, I think it is necessary to use $n > 1$ crRNA to rigorously show this result. With 3-5 more crRNAs, it will be possible to understand if this is a statistically reliable boost across different loci.

* The legend to Figure 4 should clearly describe the difference between "positive embryos" and "HDR %".

* There is a misspelling of Cpf1 in Figure 1f.

Reviewer #2 (Remarks to the Author):

Cpf1 has recently been shown to be an RNA-guided endonuclease capable to mediate site-specific genome engineering in a variety of systems. The CRISPR/Cpf1 system has a variety of features distinct from the most commonly used CRISPR/Cas9 system, which makes it an attractive option for various genome engineering applications. However, there exist conflicting reports about the robustness and efficiency of the CRISPR/Cpf1 system, with generally high levels of success reported in mammalian cells, but only low to moderate genome editing rates observed in plants and flies.

In this study Moreno-Mateos et al. implement CRISPR/Cpf1 genome engineering in zebrafish and *Xenopus*. Consistent with previous studies in ectotherm organisms they find that LbCpf1 functions as a more efficient endonuclease than AsCpf1. This gives rise to the hypothesis that the differences in AsCpf1 activity reported in the literature and found in this study are a reflection of the different temperatures at which these experiments were carried out. Consistent with this idea they demonstrate that Cpf1 activity is modulated by temperature in a range between 25-37C and that this effect is larger on AsCpf1. Lastly, the authors show that Cpf1 mediated double strand breaks are more efficiently repaired by homology directed repair utilizing an single-stranded donor template than DNA lesions created by Cas9.

I was genuinely excited by the results reported in this study. This is clearly a solid piece of work and will be of broad interest to the wider scientific community. However, there are a few issues with the manuscript in its current form that the authors should consider to revise before the manuscript can be published.

1. In general, the current manuscript is written in such a condensed style that it will be difficult for readers outside the field to follow the experiments. I strongly encourage the authors to go through the manuscript - and the results section in particular - and see where adding some additional information about the experiments will help the broad readership of Nature Communications to follow the paper.

2. The authors perform most of their experiments with pools of three crRNAs for each target gene. This should be stated more clearly in the main text. This raises the question whether the robust gene editing efficiency observed on phenotypic level in some experiments is a reflection of using crRNA pools and would not be observed when using single crRNAs. Use of sgRNA pools have been reported to substantially increase phenotypic penetrance of CRISPR/Cas9 genome engineering in mice and flies. The authors present experiments that this is not the case here (Suppl. Fig. 3f), likely because most alleles are only edited at one target site (Suppl. Data 1). This is interesting, in particular since the ability to easily multiplex crRNA in the CRISPR/Cpf1 system has been proposed as one of the potential advantages of the system. The authors should consider discussing these findings in more detail in the main text.

3. The authors suggest that temperature influences the ability of Cpf1 to access genomic target sites. In Fig. 3a it is implied that this is at least partially based on the chromatin state at the target site. This interpretation is not consistent with the finding that temperature has a big effect on the ability of Cpf1 to cut a protein-free PCR product. The text should be revised accordingly.

4. It is crucial to exclude that the higher HDR rate reported in Fig.4 is simply a reflection of a higher cutting efficiency of Cpf1 this site compared to Cas9. This result is presented in Suppl. Fig. 8a, but should also be discussed in the main text.

5. It should be made clearer in the text that the activity differences of LbCpf1 and AsCpf1 in ectotherm organisms have been already observed previously in the studies by Hu et al. (Ref. 7) and Kim et al. (Ref. 8).

6. Has the germline transmission rates been determined for embryos injected with AsCpf1 RNPs? If so, they should be included.

7. Results, second page, first paragraph, last sentence: Clarify that this experiment tested cutting efficiency in vitro.

8. Results, second page, second paragraph, second sentence: Clarify that this effect was only observed for tyr.

9. Discussion, first page, last sentence: Longer deletions by Cpf1 than typically observed for Cas9 are also reported in Port and Bullock (Ref. 9).

10. Another difference between Cpf1 and Cas9 is the type of DNA lesion they create. This might also play a role in the relative frequency with which the different DNA repair pathways are utilized in the cell. The authors might want to add this speculation to the relevant section.

General comments:

- Next time please provide line numbers, they make my life so much easier.
- I would like to thank the authors that they have chosen to accelerate science by posting a preprint and making their findings widely accessible by choosing a high-quality open access journal.

Dear Reviewers,

We would like to thank you for your insightful comments on the manuscript. We have addressed your suggestions that were mainly focused on providing more details on the result section, test off-targets and additional loci for HDR. All changes are highlighted in red. We have followed Nature Communications guidelines and corrected the title. Please find below a summary of the points raised and addressed:

Point by point answer to the reviewers:

Reviewer #1 (Remarks to the Author):

Moreno-Mateos et al. compare nuclease efficiency with LbCpf1 and AsCpf1 in zebrafish and find that brief pulses of elevated temperature enhance KO efficiency. They also compare Cpf1 with SpCas9 and find increased ssODN-based HR using Cpf1. Overall, I think these optimized parameters for efficient editing in zebrafish will be broadly useful to the community but, to ensure that this is the case, a few issues need to be addressed

Comments:

Since the authors are advocating use of Cpf1 instead of Cas9 (which works very well in zebrafish), it is necessary to compare the new CRISPRs with Cas9. For Fig. 1 and 2, the authors should provide parallel experiments with Cas9 using the same conditions/temperature changes. Does LbCpf1 outperform Cas9 for routine KO?

We appreciate the reviewer's comment and suggestion. We propose LbCpf1 expands genome editing capabilities based on a distinct PAM. In this study, we optimized the CRISPR-Cpf1 system and show that its PAM is found more often in the zebrafish genome globally increasing the number of potential target sites especially in the UTR and intergenic regions (**Supp. Fig. 1a**). Therefore, we aim to broaden the genome engineering toolkit; we now mention this in the discussion (lines 192-193).

While it is difficult to explicitly compare activity of SpCas9 and LbCpf1 for a specific locus, since they recognize different PAM sequences and they can have different nucleotide preferences for targeting (Moreno-Mateos et al. 2015; Doench et al., 2016; Kim et al. 2017), we have compared CRISPR-SpCas9 and CRISPR-LbCpf1 RNP complexes targeting the *albino* locus using three sgRNAs individually and pooled (**Fig. 1e, Supp. Fig. 3d and f and Supp. Fig. 7c and e**). SpCas9 showed a slightly stronger activity than LbCpf1 when the experiment was conducted at 28°C. However, a mere 4 h incubation at 34°C increases LbCpf1 activity to similar or even stronger levels than SpCas9. In contrast, SpCas9 activity was not modulated by higher temperatures either *in vitro* or *in vivo* (**Supp. Fig. 7**), suggesting that temperature specifically influences Cpf1 function. In addition, we compared the activity of a sgRNA and a crRNA targeting the same exon of *tyr*, *ntla* and *golden* loci at 28°C (**Supp. Fig. 9c and and Supp. Fig. 10c and h**). LbCpf1

outperformed SpCas9 targeting *tyr*, showed similar activity targeting *golden*, and exhibited slightly lower efficiency when targeting *ntla*.

In summary, our study allows now the use of Cpf1 in ectothermic animals, which was not possible before the optimization we developed. Comparing crRNAs/sgRNAs (n=4 loci using 6 crRNAs and 6 sgRNAs), both systems are very efficient and provide two comparable methods to induce indel mutations, where Cpf1 dramatically expand the target space in the genome. Yet, as shown below, the overall efficiency of induced homologous recombination is higher than Cas9 when using LbCpf1 in zebrafish (n=4 loci).

What about off-target effects? Is Cpf1 superior to Cas9 or similar in this regard (i.e. number of off-target modifications)? Even if there is greater on-target modification with Cpf1, perhaps there is greater off-target modification also and this would recommend against Cpf1 (vs. Cas9).

To avoid potential off-target effects, we only used crRNAs that do not have any predicted off-target with 2 or less mismatches in comparison with the target (Kim et al., 2017 Nat. Methods). Since higher on-target activities can also generate potential off-targets, as the reviewer commented, we have selected the 2 most efficient crRNAs used in our study to analyze their potential activities in off-targets with 3 or more mismatches. Activity on these potential off-targets with more than 3 mismatches was already detected much more frequently in SpCas9 than LbCpf1 in human cells in genome wide analysis (Tsai et al., 2015 Nat. Biotech., Kleinstiver et al., 2016 Nat. Biotech., both Guide-seq analysis).

In this revised study, we have measured the percentage of indel mutation induced by these two highly efficient crRNAs in their corresponding targets and in 13 potential off-targets using a modified version of our MutSeq pipeline (Vejnar et al., 2016 Cold Spring Harbor Protocols) (**See Methods and Supp. Fig. 5**). While we detected 84% and 95% of indel mutations in the targets, no off target activity was detected by deep sequencing in agreement with other studies in mammalian cells such as:

- 1) Kim et al 2016 Nat. Biotech. (doi: 10.1038/nbt.3609.) genome wide analysis (Digenome-seq).
- 2) Kleinstiver et al., 2016 Nat. Biotech. (doi: 10.1038/nbt.3620) genome wide analysis (Guide seq).
- 3) Kim et al. 2016. Nat. Biotech. (doi: 10.1038/nbt.3614.) and Hur et al., 2016 Nat. Biotech. (doi: 10.1038/nbt.3596.) No/low off target, whole genome sequencing and looking for potential off-targets, respectively (sequence based).
- 4) Kim et al., 2017 Nat. Methods (doi: 10.1038/nmeth.4104): In vivo high-throughput profiling of Cpf1 activity (on and off target activity by deep sequencing).

All these studies indicate that CRISPR–Cpf1 has equal or higher targeting specificity than CRISPR–Cas9.

All together, our study is consistent with previous studies shown above and show that highly efficient crRNAs are also specific.

The PAM for Cpf1 has now been shown to be TTTV not TTTN. Did the authors use TTTT sequences and, if so, can they comment on differences with TTTV PAMs?

The referee is correct. Following the reviewer's suggestion and based on the genome wide analysis in mammalian cells, we have changed the PAM sequence in the paper to TTTV, and we have corrected the **Supp. Fig 1a** and revised the PAM sequence in our computational tool, CRISPRscan, accordingly.

Interestingly, when we selected the crRNA targets sites, the article demonstrating lower on-target efficiency for the TTTT PAM in mammalian cells had not been published yet (Kim et al, 2017 Nat Methods doi: 10.1038/nmeth.4104). In any case, only one crRNA used in our study has a TTTT PAM (alb 1 for *X. tropicalis* **Supp Table 1**). We evaluated the activity of this crRNA in combination with other 2 crRNAs (alb 2 and alb 3 for *X. tropicalis*, **Supp Table 1**) and interestingly we did detect mutagenesis activity for this crRNA (**Supp Data 1**). Despite the lower predicted activity, this crRNA induced on-target indels. Therefore, while we cannot comment on the genome-wide efficiency of the TTTT PAM, we show that it is capable of inducing on-target mutations in *X. tropicalis*. This is consistent with previously published results. Kim *et al.* 2017 (doi: 10.1038/nmeth.4104) showed that LbCpf1 activity is not completely abolished when TTTT is the PAM. Thus, it is plausible to find some activity associated to this type of target. We have now added a note about this in Supplementary Table 1.

Although Figure 4 looks very promising, I think it is necessary to use $n > 1$ crRNA to rigorously show this result. With 3-5 more crRNAs, it will be possible to understand if this is a statistically reliable boost across different loci.

We thank the reviewer for his/her comment and apologize for the misunderstanding. We had originally provided two loci, and we have now expanded our analysis to 4 loci in total.

First, we identify the conditions that optimize LbCpf1 activity with specific ssDNA oligos to induce HDR: single-strand donor DNA complementary to the target strand and with longer homology in the PAM-proximal arm (**Fig. 4, Supp. Fig. 9 and 10**). This is consistent across different loci and is different from the optimized conditions for Cas9 (Richardson et al 2016. Nat Biotech.).

Second, we show that LbCpf1 is more efficient (up to 4-fold) in homology-directed repair in 2 out 4 loci tested (**Fig. 4, Supp. Fig. 9 and 10**), and has a similar efficiency to SpCas9 in the other 2 loci (**Supp Fig. 10**). In summary, the efficiency of induced homologous recombination is either equal or higher than SpCas9 when using LbCpf1 in zebrafish (n=4), suggesting this tool can

be used not only as a complement to SpCas9 but also as an improved alternative for HDR approaches using ssDNA oligos.

The legend to Figure 4 should clearly describe the difference between “positive embryos” and “HDR %”.

We apologize for the lack of clarity. We have clarified in the legends where % of HDR was shown (**Fig. 4, Supp. Fig. 9 and 10**) and corrected the figure accordingly:

% of HDR: amount of integrated DNA per total amount of genomic DNA per embryo (see Methods for details).

Positive Embryos: number of embryos per condition showing a detectable qPCR amplification signal.

There is a misspelling of Cpf1 in Figure 1f.

We apologize for this typo. We have corrected it.

Reviewer #2 (Remarks to the Author):

Cpf1 has recently been shown to be an RNA-guided endonuclease capable to mediate site-specific genome engineering in a variety of systems. The CRISPR/Cpf1 system has a variety of features distinct from the most commonly used CRISPR/Cas9 system, which makes it an attractive option for various genome engineering applications. However, there exist conflicting reports about the robustness and efficiency of the CRISPR/Cpf1 system, with generally high levels of success reported in mammalian cells, but only low to moderate genome editing rates observed in plants and flies.

In this study Moreno-Mateos et al. implement CRISPR/Cpf1 genome engineering in zebrafish and Xenopus. Consistent with previous studies in ectotherm organisms they find that LbCpf1 functions as a more efficient endonuclease than AsCpf1. This gives rise to the hypothesis that the differences in AsCpf1 activity reported in the literature and found in this study are a reflection of the different temperatures at which these experiments were carried out. Consistent with this idea they demonstrate that Cpf1 activity is modulated by temperature in a range between 25-37C and that this effect is larger on AsCpf1. Lastly, the authors show that Cpf1 mediated double strand breaks are more efficiently repaired by homology directed repair utilizing an single-stranded donor template than DNA lesions created by Cas9.

I was genuinely excited by the results reported in this study. This is clearly a solid piece of work and will be of broad interest to the wider scientific community. However, there are a few issues with the manuscript in its current

form that the authors should consider to revise before the manuscript can be published.

1. In general, the current manuscript is written in such a condensed style that it will be difficult for readers outside the field to follow the experiments. I strongly encourage the authors to go through the manuscript - and the results section in particular - and see where adding some additional information about the experiments will help the broad readership of Nature Communications to follow the paper.

We thank the reviewer for their enthusiasm and apologize for the lack of clarity in the manuscript. We have reviewed the manuscript, particularly the results section, to provide more information about the experiments. We hope that these changes improve the clarity of the manuscript.

2. The authors perform most of their experiments with pools of three crRNAs for each target gene. This should be stated more clearly in the main text. This raises the question whether the robust gene editing efficiency observed on phenotypic level in some experiments is a reflection of using crRNA pools and would not be observed when using single crRNAs. Use of sgRNA pools have been reported to substantially increase phenotypic penetrance of CRISPR/Cas9 genome engineering in mice and flies. The authors present experiments that this is not the case here (Suppl. Fig. 3f), likely because most alleles are only edited at one target site (Suppl. Data 1). This is interesting, in particular since the ability to easily multiplex crRNA in the CRISPR/Cpf1 system has been proposed as one of the potential advantages of the system. The authors should consider discussing these findings in more detail in the main text.

We appreciate this observation and the reviewer's comment. We have described the use of pools of crRNAs more clearly in the text (lines 78,100). Initially, we used pools of crRNAs because we thought that combining crRNAs could increase the generation of loss-of-function mutants, in the absence of a detailed analysis of crRNA efficiency for each locus. However, we later observed that using a pool of crRNAs was not significantly more efficient than the most efficient crRNA in that pool (**Supp. Fig. 3f**) (Chi Square p-value=0.0585 comparing alb 2 vs pool, Chi-square test). In addition, when we used only one crRNA targeting *tyr*, we observed a similar activity than when we used it in combination with other two crRNAs (**Supp. Fig 9c vs Fig 1g-10 fml-**), suggesting that the activity of the pool is driven by this crRNA. All together, these observations suggest that using LbCpf1-crRNA RNP complexes *in vivo*, a) there is a differential activity between different crRNAs as shown in Kim et al (Nat Methods, 2017) in human cells, and b) the activity of a pool may be mainly driven by the most efficient crRNA in that pool. We have now added this to the results section (lines 94-98) and in the Supp. Fig 3f legend.

3. The authors suggest that temperature influences the ability of Cpf1 to access genomic target sites. In Fig. 3a it is implied that this is at least partially based on the chromatin state at the target site. This interpretation is not consistent with the finding that temperature has a big effect on the ability of Cpf1 to cut a protein-free PCR product. The text should be revised accordingly.

We would like to thank the reviewer for this insightful observation. There seem to be several factors that converge in the increased efficiency of Cpf1 at higher temperature. We indeed observe an increase in the ability of Cpf1 to access genomic target sites by using CRISPR-proxy, but while the lower ability of Cpf1 to access genomic DNA is likely one of the reasons for the decreased Cpf1 activity at temperatures below 37°C, we cannot rule out other effects because *in vitro* activity is also lower. This is particularly true for AsCpf1. We have now mentioned this in the discussion (lines 203-206).

4. It is crucial to exclude that the higher HDR rate reported in Fig. 4 is simply a reflection of a higher cutting efficiency of Cpf1 this site compared to Cas9. This result is presented in Suppl. Fig. 8a, but should also be discussed in the main text.

We agree with the reviewer, and indeed we carefully monitored the activity of each sgRNA and crRNA to make sure their cleavage activity was as close as possible. The activity for SpCas9 and LbCpf1 using these particular sgRNAs and crRNA in **Fig 4** is included in **Supp. Fig. 9a**, and it shows that Cas9 activity is slightly higher than LbCpf1. As the rate of HDR in LbCpf1 is higher while cutting efficiency is lower, the results observed in Fig. 4 are consistent with a great HDR efficiency in LbCpf1 compared to SpCas9. We show a consistent result in Supp. Fig. 8 and, although here Cas9 activity is slightly lower than Cpf1 (**Supp. Fig 9c**), the differences in HDR are so large, that it is unlikely that these differences are driven by the small differences in crRNA activity.

We have now used two more loci (*ntla* and *golden*) to measure HDR. In these two cases, LbCpf1 and SpCas9 showed similar efficiency inducing HDR. Interestingly, in *ntla* experiment SpCas9 showed slightly higher activity, but it did not trigger higher HDR for SpCas9. All together, these results suggest that LbCpf1 activity induces equal or more HDR than SpCas9 and that it is not solely driven by the efficiency of the crRNA/gRNA.

5. It should be made clearer in the text that the activity differences of LbCpf1 and AsCpf1 in ectotherm organisms have been already observed previously in the studies by Hu et al. (Ref. 7) and Kim et al. (Ref. 8).

We thank the reviewer for this suggestion. We have now mentioned it in the **discussion section**, remarking that now our study provide a context to

understand how AsCpf1 activity is reduced in ectothermic organisms: lines 199-200

6. Has the germline transmission rates be determined for embryos injected with AsCpf1 RNPs? If so, they should be included.

We have crossed F0 injected animals and determined the rate of germline transmission for fish injected with AsCpf1 RNP complexes targeting *albino* and *tyrosinase*. Consistent with their differential activity, the germ line transmission of AsCpf1 was much more lower than for LbCpf1, and we have now included these results in **Supplementary Fig. 3g,h**.

7. Results, second page, first paragraph, last sentence: Clarify that this experiment tested cutting efficiency in vitro.

We have now clarified it, indicating that this experiment was performed *in vitro* (line 122).

8. Results, second page, second paragraph, second sentence: Clarify that this effect was only observed for tyr.

We observed a temperature effect also in albino (*slc45a2*) but only when pre-crRNAs were used (Supplementary Fig. 6b), most likely due to a slightly higher activity that it shows up when the embryos are incubated at 34°C. We have now corrected it: “We observed that a brief incubation of embryos at 34°C post-injection significantly increased AsCpf1-mediated gene editing of *tyr* and *slc45a2* when using pre-crRNAs (Fig. 2a-c, Supplementary Fig. 6b)” (lines 126-128).

9. Discussion, first page, last sentence: Longer deletions by Cpf1 than typically observed for Cas9 are also reported in Port and Bullock (Ref. 9).

We have added the reference.

10. Another difference between Cpf1 and Cas9 is the type of DNA lesion they create. This might also play a role in the relative frequency with which the different DNA repair pathways are utilized in the cell. The authors might want to add this speculation to the relevant section.

We thank the referee for this useful comment. We have mentioned this possibility in the discussion section supported by a study in mammalian cells:

“It is also possible that the different DNA lesion induced by Cpf1 and Cas9 affects the repair process (Bothmer A et al., 2017)” (lines 216-218).

General comments:

Next time please provide line numbers, they make my life so much easier.

We apologize for this lack of clarity. We have added it now for the revision.

I would like to thank the authors that they have chosen to accelerate science by posting a preprint and making their findings widely accessible by choosing a high-quality open access journal.

We thank the reviewer for this comment. We also feel very enthusiastic about preprints and the feedback we obtained from the community making these results available pre-publication. This has already helped the scientific community since, based on our results, the temperature-controlled activity for AsCpf1 has been recently validated in *Drosophila*
<http://www.crisprflydesign.org/1036-2/>

REVIEWERS' COMMENTS:

Reviewer #1 (Remarks to the Author):

The authors have addressed my concerns. In particular, I appreciate their efforts to add more data to the HR figure.

Reviewer #2 (Remarks to the Author):

The revised manuscript has been significantly improved over the initial version. The authors now provide targeted deep sequencing at potential off-target sites that point towards a high specificity of Cpf1. This is consistent with several reports in the recent literature. Furthermore, the authors have expanded their study of Cpf1 mediated HDR to additional target sites. In addition, they have made multiple changes to the text, which improved the clarity of the manuscript. I strongly recommend publication of this paper.

REVIEWERS' COMMENTS:

Reviewer #1 (Remarks to the Author):

The authors have addressed my concerns. In particular, I appreciate their efforts to add more data to the HR figure.

Reviewer #2 (Remarks to the Author):

The revised manuscript has been significantly improved over the initial version. The authors now provide targeted deep sequencing at potential off-target sites that point towards a high specificity of Cpf1. This is consistent with several reports in the recent literature. Furthermore, the authors have expanded their study of Cpf1 mediated HDR to additional target sites. In addition, they have made multiple changes to the text, which improved the clarity of the manuscript. I strongly recommend publication of this paper.

We thank the reviewers for their comments